# Engineered long-acting Irisin-albumin binding domain fusion protein for enhanced anti-inflammatory efficacy in lipopolysaccharide-induced systemic inflammation

Jicun Zhu[1,9], Yujie Zhang[2,3,9], Xinwei Wang[2,9], Lei Peng[2], Xiaojia Ma[2], Zan Qiu[2], Zirui Kang[2], Fangyuan Zheng[2], Xiaoyu Zhang[4], Mengyuan Song[4], Jia Du[5], Yuan Shi[6], Lie Yu ⓘ [7] ✉, Chenxi Gu ⓘ [8] ✉ & Jianxiang Shi ⓘ [2] ✉

Irisin is a peptide hormone with notable anti-inflammatory and metabolic regulatory effects but has limited clinical utility due to its extremely short plasma half-life (< 1 h). In this study, we engineered an albumin-binding domain (ABD)-conjugated Irisin (ABD-Irisin) fusion protein to significantly extend half-life and enhance therapeutic potency. ABD-Irisin fusion protein was successfully expressed in HEK-293F cells, purified, and validated through functional assays including lipid droplet reduction and western blot assays. Pharmacokinetic studies demonstrated that ABD-Irisin markedly prolonged the plasma half-life of Irisin to approximately 10 h, substantially surpassing native Irisin, and showed enhanced tissue distribution in vivo. In a lipopolysaccharide (LPS) induced mouse model of systemic inflammation, both Irisin and ABD-Irisin significantly reduced plasma TNF-α levels, splenomegaly, and histopathological inflammation. Notably, ABD-Irisin (500 µg/kg) demonstrated significantly enhanced suppression of plasma IL-6 and splenic inflammatory cytokines (IL-1β, IL-10) compared to native Irisin. Single-cell RNA sequencing further revealed that ABD-Irisin robustly suppressed the activation of the TLR4-MyD88-NF-κB signaling axis in bone marrow immune cells, outperforming unmodified Irisin. These findings demonstrate that ABD conjugation is an effective strategy to enhance the pharmacokinetics and anti-inflammatory efficacy of Irisin, highlighting ABD-Irisin as a promising therapeutic candidate for inflammatory diseases.

Irisin is a peptide hormone encoded by the *FNDC5* gene and was first discovered by Boström *et al.* in 2012[1]. Irisin is primarily produced by skeletal muscle and adipose tissue, and its circulating level are positively correlated with exercise intensity and frequency[2,3]. Irisin exerts its physiological effects by binding to receptors in the αV integrin complex family[4], subsequently activating multiple signaling pathways such as AMPK-UCP2, MAPK, and PI3K/AKT[5]. These pathways are involved in critical metabolic processes, including energy homeostasis and adipose tissue metabolism. Irisin

[1]Department of Pharmacy, The First Affiliated Hospital of Zhengzhou University, Zhengzhou, China. [2]Henan Institute of Medical and Pharmaceutical Sciences, Zhengzhou University, Zhengzhou, China. [3]Henan Key Laboratory of Microbiome and Esophageal Cancer Prevention and Treatment & Henan Key Laboratory of Cancer Epigenetics, The First Affiliated Hospital (College of Clinical Medicine) of Henan University of Science and Technology, Luoyang, China. [4]School of Basic Medical Sciences, Zhengzhou University, Zhengzhou, China. [5]College of Public Health, Zhengzhou University, Zhengzhou, China. [6]Anyang Tumor Hospital, The Affiliated Anyang Tumor Hospital of Henan University of Science and Technology, Anyang, China. [7]Department of Neurology, The First Affiliated Hospital of Zhengzhou University, Zhengzhou, China. [8]Department of Orthopaedic Surgery, The First Affiliated Hospital of Zhengzhou University, Zhengzhou, China. [9]These authors contributed equally: Jicun Zhu, Yujie Zhang, Xinwei Wang. ✉e-mail: fccyul@zzu.edu.cn; guchenxi@zzu.edu.cn; jianxiangshi@zzu.edu.cn

promotes the browning of subcutaneous white adipose tissue, thereby enhancing energy expenditure and thermogenesis[6,7]. Notably, Irisin modulates inflammatory responses by reducing pro-inflammatory cytokines (TNF-α, IL-6) and elevating anti-inflammatory cytokines (IL-10), thus facilitating tissue repair and regeneration[8].

Despite these beneficial effects, the clinical application of Irisin is limited due to its extremely short plasma half-life (<1 h), necessitating frequent dosing that could elevate treatment costs and increase potential adverse effects[9]. Consequently, developing strategies to extend the half-life and enhance the efficacy of Irisin remains a great challenge.

Fusion protein technology has emerged as an effective approach to address these pharmacokinetic limitations by coupling biologically active proteins with long-circulating carriers such as immunoglobulins, albumin, or transferrin[10]. Among these carriers, human serum albumin (HSA) is particularly advantageous due to its biocompatibility, minimal immunogenicity, and notably long serum half-life (approximately 19 days)[11]. The albumin-binding domain (ABD), derived from streptococcal protein G, binds non-covalently and with high affinity to both human and mouse serum albumin, extending the circulation time of various therapeutic agents[12,13]. The ABD fusion strategy has successfully prolonged the half-life of several drugs, including glucagon-like peptide-1, adalimumab Fab fragments, and single-domain antibody-drug conjugates, thereby significantly enhancing their therapeutic efficacy in diabetes, autoimmune diseases, and oncology[14–16]. However, the potential of ABD conjugation to enhance Irisin's pharmacokinetic profile and therapeutic efficacy remains unexplored.

In this study, we constructed, expressed, and evaluated an ABD-Irisin fusion protein aiming to overcome Irisin's short half-life limitations and enhance its anti-inflammatory effects. Our specific objectives included: 1) validating whether ABD fusion can substantially prolong Irisin's half-life; 2) comparing the anti-inflammatory efficacy of ABD-Irisin versus Irisin in an LPS-induced mouse inflammation model; and 3) elucidating the underlying molecular mechanism through single-cell RNA sequencing (scRNA-seq) analyses. Our findings provide critical theoretical insights and experimental evidence supporting ABD-Irisin as a promising long-acting therapeutic candidate for treating inflammatory diseases.

## Results

### Expression and purification of Irisin and ABD-Irisin proteins
Irisin and ABD-Irisin proteins were engineered with CD5 signal peptide followed by an N-terminal His-tag separated by a TEV cleavage site (Fig. S1). The proteins were expressed in HEK 293 F cells as secreted proteins and supernatants were collected to purify Irisin and ABD-Irisin, respectively. Affinity chromatography using Ni-NTA agarose effectively purified Irisin and ABD-Irisin proteins. Both proteins were cleaned with 50 mM imidazole and eluted with 200 mM imidazole (Fig. S2). SDS-PAGE analysis confirmed the molecular weights of Irisin and ABD-Irisin to be approximately 20-32 kDa and 30-32 kDa, respectively (Fig. 1A). Western blot using anti-Irisin and anti-His antibodies verified that proteins of interest were successfully expressed and purified (Fig. 1B, C). Identity and purity (> 95%) were validated for both proteins.

### Structure and activity of Irisin and ABD-Irisin proteins
In the Input assay, HSA was detected *via* immunoblotting in HSA-containing groups. In the Pulldown assay, Ni-NTA magnetic beads immobilized Irisin, ABD-Irisin. And HSA was added to corresponding tube. Analysis of the eluates revealed that only ABD-Irisin bound HSA (Fig. S3), demonstrating that ABD-Irisin could directly bind to HSA, whereas Irisin could not.

Oil Red O staining revealed a significant reduction in lipid droplets in adipocytes treated with Irisin or ABD-Irisin compared to untreated controls (Fig. 1D). Isopropanol extraction followed by absorbance measurement quantitatively confirmed significant lipid droplet reduction at concentrations of 100, 200, and 400 ng/mL for both proteins, indicating both purified proteins preserved lipid-reducing bioactivity (Fig. 1E, F).

### Plasma half-life and tissue distribution of Irisin and ABD-Irisin proteins
Plasma concentrations and pharmacokinetic parameters for Irisin and ABD-Irisin in mice are summarized in Fig. 1G, H and Table S1. Irisin administered at doses of 100, 500, and 1000 μg/kg exhibited short half-lives ($t_{1/2}$) of 0.39, 0.3, and 0.32 h, respectively, with peak concentrations ($C_{max}$) reaching 27.05, 176.89, and 281.49 ng/mL at 0.5 h ($T_{max}$). In contrast, ABD-Irisin displayed significantly prolonged half-lives (9.6, 11.5, and 10.46 h), with higher peak concentrations (303.6, 435.57, and 669.71 ng/mL), and delayed $T_{max}$ (1 h). ABD-Irisin maintained notably higher plasma concentrations than Irisin at equivalent doses.

Immunohistochemistry conducted 2 h post-injection (500 μg/kg) demonstrated substantially stronger ABD-Irisin signals in liver, lung, and kidney tissues compared to Irisin (Fig. 2A). Irisin accumulated predominantly in lung tissue (Fig. 2B). Notably, ABD-Irisin significantly increased tissue concentrations of Irisin in liver, lung, and kidney compared to native Irisin (Fig. 2B).

### Anti-inflammatory effect of Irisin and ABD-Irisin proteins
LPS treatment significantly elevated plasma IL-6 and TNF-α concentrations compared to PBS-treated controls (Fig. 3). ABD-Irisin (500 μg/kg) showed significantly stronger IL-6 suppression than native Irisin ($P < 0.05$), though TNF-α reduction was comparable at this dose.

Baseline body weights among treatment groups showed no significant differences. LPS challenge led to progressive weight loss, while mice receiving Irisin or ABD-Irisin maintained body weights comparable to PBS controls throughout the observation period (Table S2, Fig. S4). LPS exposure also induced significant splenomegaly compared to PBS-treated mice; treatment with Irisin or ABD-Irisin notably reduced spleen enlargement, although differences between Irisin and ABD-Irisin groups were not significant (Fig. S5).

Hematoxylin and eosin (H&E) staining was performed on major organs (heart, liver, spleen, lung, and kidney) of mice from the PBS, LPS, LPS +Irisin (500 μg/kg), and LPS + ABD-Irisin (500 μg/kg) groups. H&E-stained sections revealed inflammatory cell infiltration around the central veins of hepatic lobules in the LPS group (Fig. 4A). In contrast, minimal evidence of tissue inflammation was observed in both the LPS+Irisin and LPS + ABD-Irisin groups.

Immunohistochemical analysis showed LPS-induced elevation of IL-1β and IL-10 expression in spleen tissues. Both Irisin and ABD-Irisin significantly reduced these cytokine levels, with ABD-Irisin demonstrating superior suppression (Fig. 4B, C).

### Single-cell RNA sequencing
Single-cell RNA sequencing of bone marrow samples from mice (n = 3 per group, except for LPS group, n = 2) yielded a total of 51,429 cells, including 20,724 for PBS group, 7247 for LPS group, 12,641 for LPS+Irisin group, and 10,817 for LPS + ABD-Irisin group. Unsupervised clustering analysis identified distinct immune lineages, including lymphoid and myeloid cells, and further classified them into subpopulations: B cells, common myeloid progenitors (CMPs), dendritic cells (DCs), granulocytes, monocytes/macrophages, NK cells, erythrocytes and T cells (Figure S6).

The results showed that LPS treatment significantly increased myeloid cell populations, including granulocytes and monocytes/macrophages, while reducing lymphoid populations, particularly B and T cells. Treatment with Irisin or ABD-Irisin reversed these trends, restoring immune cell balance (Figure S7).

Granulocytes from LPS-treated mice exhibited upregulated expression of pro-inflammatory genes such as *Saa3* and *Lipg*, and downregulated anti-inflammatory genes including *Zfp36* and *Btg2* (Figure S8A). KEGG pathway enrichment analysis indicated that up-regulated genes were enriched in granulocyte activation pathways, including neutrophil extracellular trap formation, JAK-STAT signaling pathway (Figure S8B). B cells showed up-regulated *Ighg3* and *Jchain*, and down-regulated *Cd79a* and *Cd79b* (Figure S8C). KEGG pathway enrichment analysis indicated that up-regulated

**Fig. 1 | Purification, western blot analysis, activity validation and pharmacokinetics of Irisin and ABD-Irisin fusion proteins. A** Protein purification. **B** Western blot using anti-His antibody confirming N-terminal His-tag in both constructs. **C** Western blot using anti-Irisin antibody verifying protein integrity. **D** Oil Red O staining of adipocytes. **E** Effects of different concentrations of Irisin and ABD-Irisin on lipid droplet formation (n = 3 per concentration group). **F** Comparison of lipid droplet formation between Irisin and ABD-Irisin at the same concentration (n = 3 per concentration group). **G** Concentration changes of Irisin protein in mice plasma (n = 4 per dose group). **H** Concentration changes of ABD-Irisin fusion protein in mice plasma (n = 4 per dose group). Data are presented as mean ± S.D. Statistical significance: *$P < 0.05$, **$P < 0.01$, ***$P < 0.001$, ****$P < 0.0001$, ns: not significant.

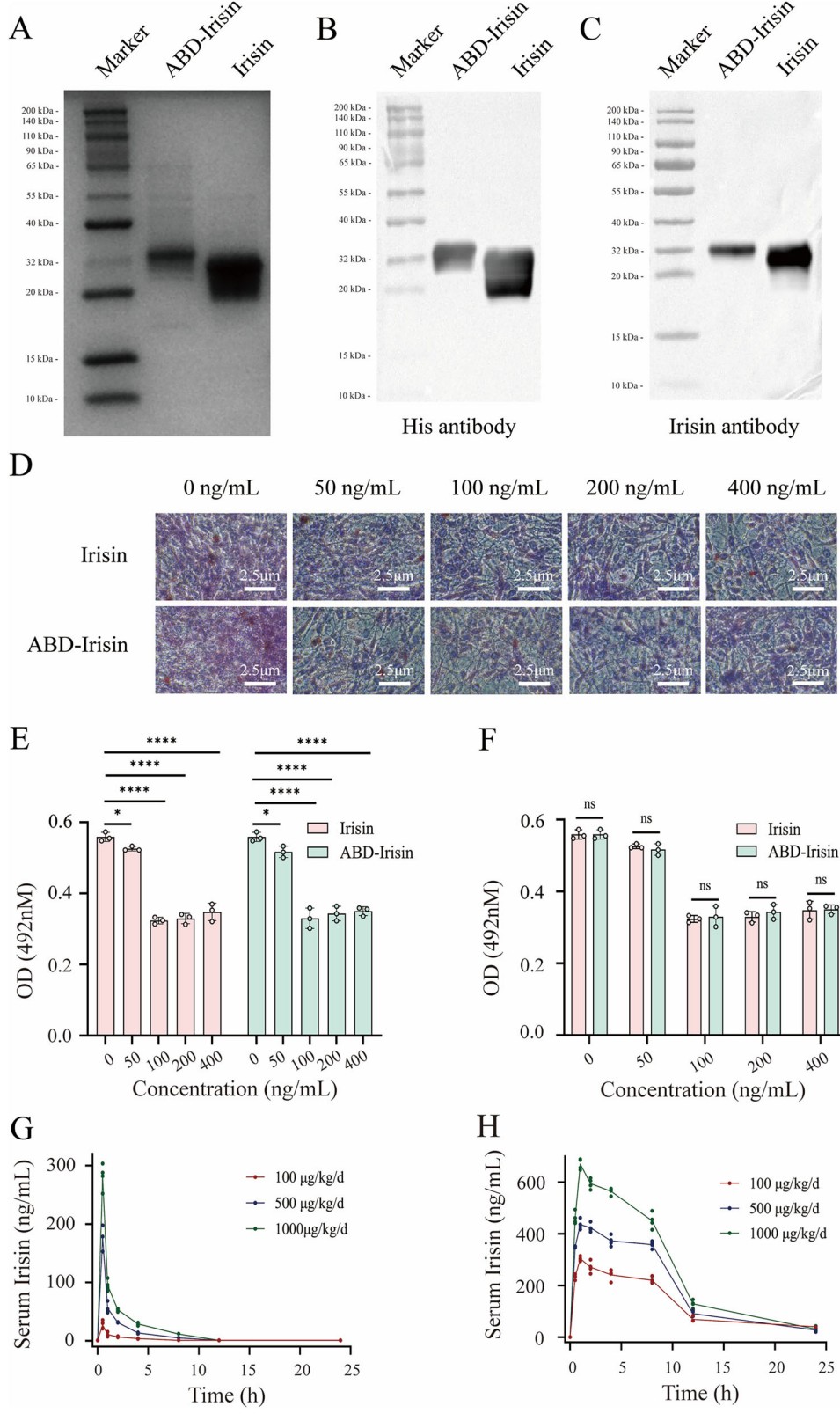

genes were enriched in protein processing in endoplasmic reticulum and various types of N-glycan biosynthesis while down-regulated genes were enriched in protein processing in B cell receptor signaling pathway, hematopoietic cell lineage (Figure S8D).

Irisin upregulated *Isg15*, *Rsad2* and *Ccl6*, and down-regulated *Stfa1* and *Staf2* in granulocytes (Figure S9A). KEGG pathway enrichment analysis in

granulocytes showed that upregulated genes were enriched in antigen processing and presentation while down-regulated genes were enriched in JAK-STAT signaling pathway and TNF signaling pathway (Figure S9B). B cells showed up-regulated *Cd79a*, *Cd74* and *Spib*, and down-regulated *Jchain* and *Iglv1* (Figure S9C). KEGG pathway enrichment analysis indicated that up-regulated genes were enriched in B cell receptor signaling

**Fig. 2 | Immunohistochemical staining of Irisin antibody in tissues of mice treated with Irisin and ABD-Irisin. A** Immunohistochemical staining images of different organ tissues. **B** Differences in mean optical density of immunohistochemical staining across organs (n = 7 per group). Data are presented as mean ± S.D. Statistical significance: $*P < 0.05, **P < 0.01, ***P < 0.001, ****P < 0.0001$, ns: not significant.

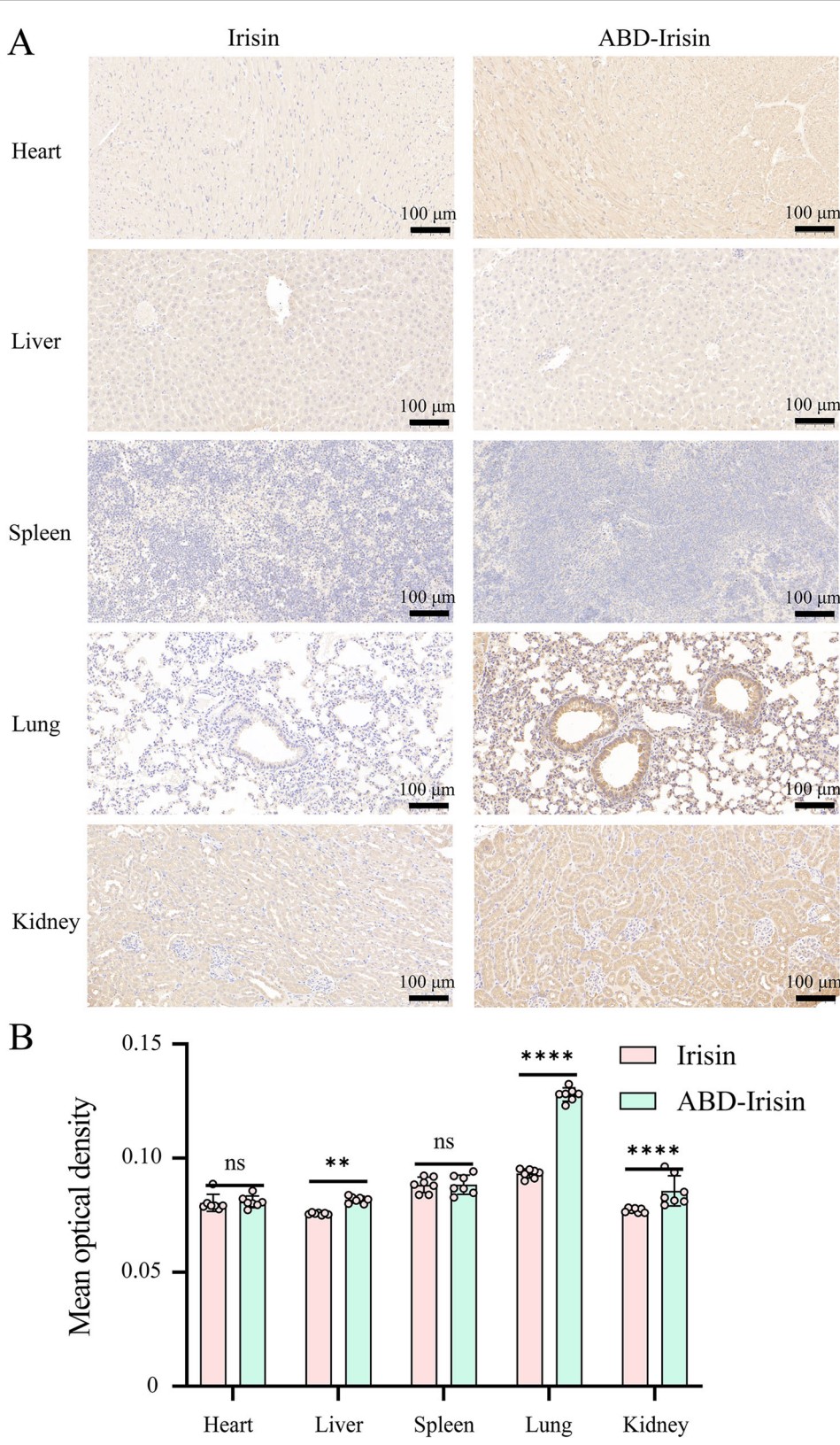

pathway and antigen processing and presentation while down-regulated genes were enriched in protein processing in endoplasmic reticulum, protein export, and N-Glycan biosynthesis (Figure S9D).

Compared to Irisin, ABD-Irisin treatment significantly down-regulated pro-inflammatory genes including *Csf3r, Cxcr2, S100a8, Il1b and Isg15* in granulocytes (Fig. 5A). KEGG pathway enrichment analysis revealed that up-regulated genes were enriched in cAMP signaling pathway and efferocytosis, and down-regulated genes were enriched in Fc gamma R-mediated phagocytosis, JAK-STAT signaling pathway, neutrophil extracellular trap formation, and TNF signaling pathway (Fig. 5B). ABD-Irisin treatment up-regulated *Slpi* and *Jchain*, and down-regulated interferon-stimulated genes, including *Cd79a, Rpl18a* and *Rps27* in B cells

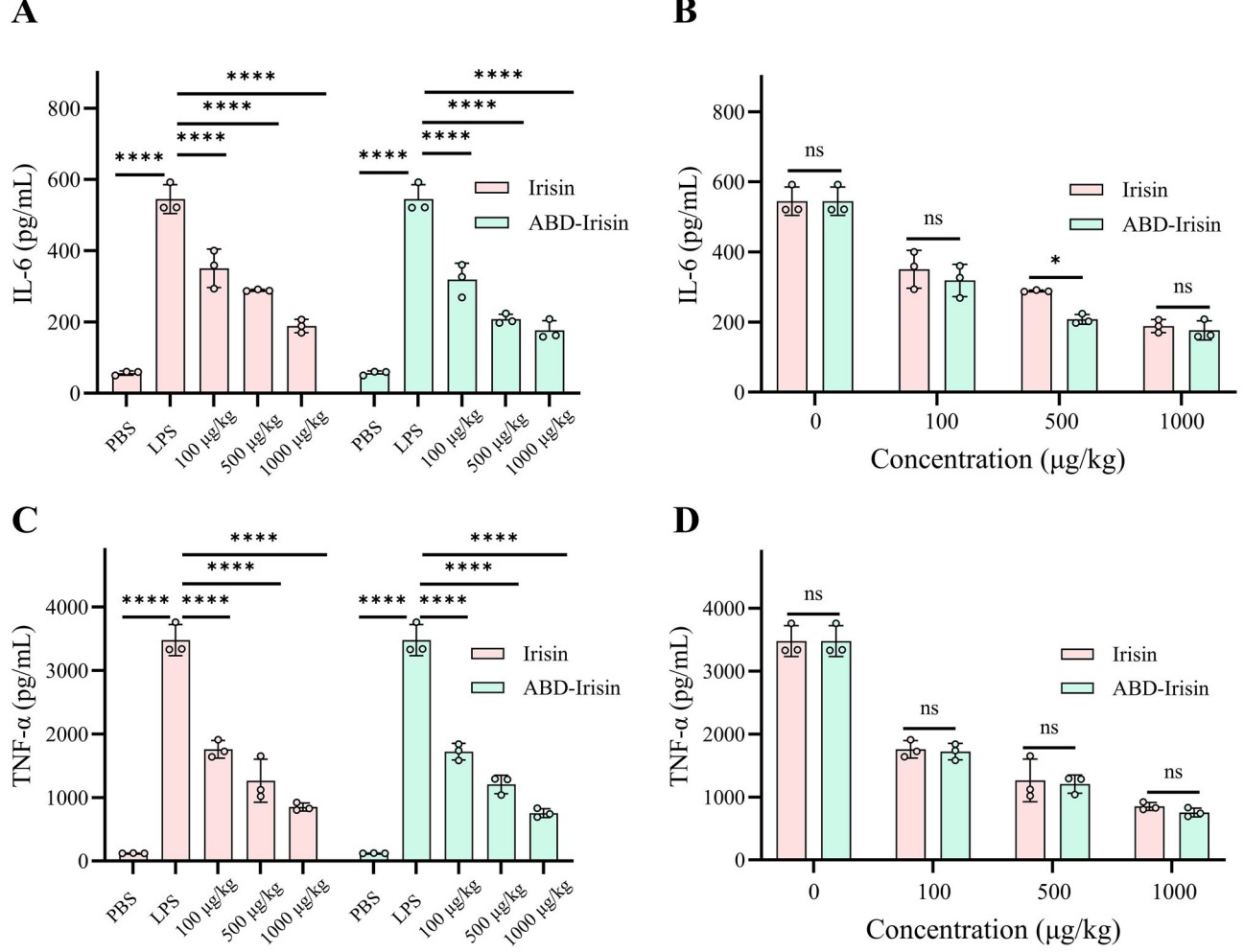

**Fig. 3 | Plasma concentrations of IL-6 and TNF-α in peripheral blood of mice from different treatment groups (n = 3 per group). A** IL-6 concentration differences among treatment groups. **B** IL-6 concentration comparison between Irisin and ABD-Irisin at the same concentration. **C** TNF-α concentration differences among treatment groups. **D** TNF-α concentration comparison between Irisin and ABD-Irisin at the same concentration. Data are presented as mean ± S.D. Statistical significance: *$P < 0.05$, **$P < 0.01$, ***$P < 0.001$, ****$P < 0.0001$, ns: not significant.

compared with Irisin treatment group (Fig. 5C). KEGG pathway enrichment analysis revealed that up-regulated genes were enriched in protein processing in endoplasmic reticulum and protein export, and downregulated genes were enriched in antigen processing and presentation, B cell receptor signaling pathway, JAK-STAT, NK-κB and Toll-like receptor signaling pathway (Fig. 5D).

The TLR4-MyD88 pathway plays a crucial role in mediating LPS-induced inflammation. In current study, LPS upregulated the expression of *Lbp*, *Cd14*, *Ly96*, *Tlr4* and *Myd88*, which are key components in TLR4-MyD88 pathway, while Irisin and ABD-Irisin suppressed both gene expression and downstream signaling, as evidenced by reduced expression of TLR4-MD2 pathway related genes (Fig. 6). Irisin and ABD-Irisin also reversed LPS-induced NF-κB activation by modulating key regulatory genes, such as *Ikbkg*, *Nfkbia*, *Nfkbib*, *Nfkb1*, *Chuk*, *Ikbkb*, *Nfkbie*, *Nfkb2* and *Rela*. ABD-Irisin exhibited a more pronounced inhibitory effect (Fig. 7).

IHC results in spleen tissues validated that protein level of *Lbp*, *Cd14*, *Tlr4*, *Myd88* and *NF-κB* in TLR4-MyD88-NF-κB signaling pathway significantly reduced in Irisin and ABD-Irisin treated groups compared with LPS group (Fig. 8). This is consistent with scRNA-seq findings.

## Discussion

In this study, we successfully designed a novel long-acting fusion protein, ABD-Irisin, by conjugating Irisin with an albumin-binding domain. Our findings clearly demonstrate that ABD-Irisin significantly extends the plasma half-life and enhances the anti-inflammatory potency of Irisin. This study provides critical theoretical insights and strong experimental evidence supporting ABD-Irisin as a promising therapeutic agent for inflammatory diseases.

The increase in molecular weight resulting from ABD conjugation likely contributes to reduced renal filtration and clearance[17]. By binding to albumin, ABD-Irisin also benefits from enhanced tissue uptake, delayed metabolic processing, and sustained plasma concentrations, significantly prolonging its effective half-life and bioavailability. Moreover, ABD-Irisin avoids lysosomal degradation through FcRn-mediated recycling and transcytosis pathways[18]. Notably, the half-life extension achieved with ABD-Irisin (approximately 10 h) represents a substantial improvement compared to native Irisin, which typically has a half-life of less than 1 h. Compared to other approaches such as PEGylation, ABD-Irisin offers superior biocompatibility and reduced immunogenicity, as it binds non-covalently to serum albumin, thereby avoiding potential impairment of function or immune responses associated with chemical modifications[19].

Irisin, a metabolic regulator, is widely recognized for its role in mitigating obesity and metabolic dysfunction[20]. Irisin suppresses lipid accumulation through activation of WNT and ERK pathways, thereby enhancing energy expenditure and metabolic enzyme expression in

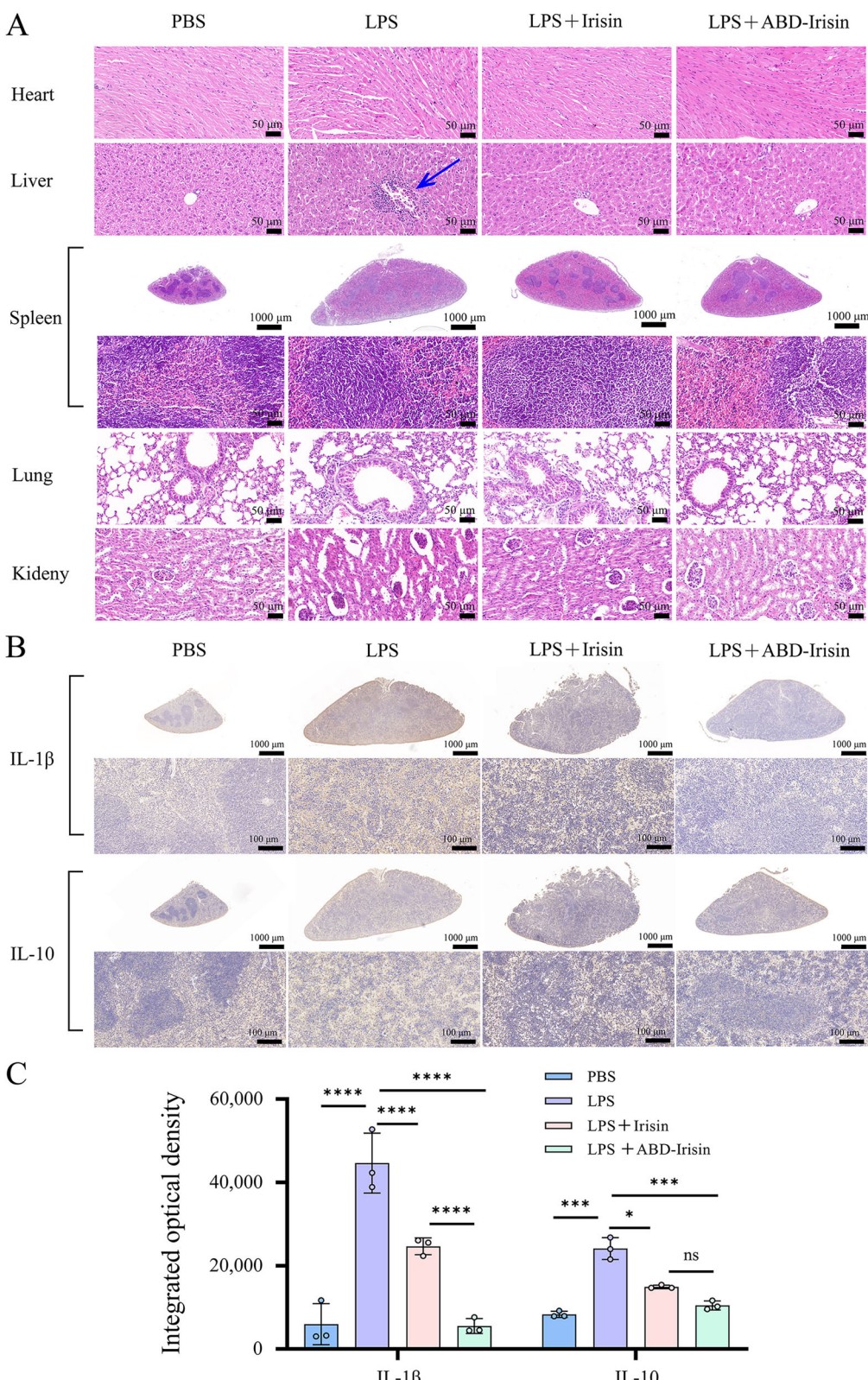

**Fig. 4 | H&E staining and immunohistochemical analysis of tissues from mice in different treatment groups. A** H&E staining of heart, liver, spleen, lung, and kidney tissues from the PBS, LPS, LPS+Irisin (500 µg/kg), and LPS + ABD-Irisin (500 µg/kg) groups. Samples from the five tissues were fixed using 4% paraformaldehyde for 24 h at room temperature, and embedded using paraffin. Blue arrows indicate inflammatory cell infiltration around the central veins of hepatic lobules in the LPS group. **B** Immunohistochemical staining of IL-1β and IL-10 in spleen tissues. **C** Differences in cumulative optical density of immunohistochemical staining (IL-1β and IL-10) in spleen tissues among treatment groups (n = 3 per group). Data are presented as mean ± S.D. Statistical significance: *P < 0.05, **P < 0.01, ***P < 0.001, ****P < 0.0001, ns: not significant.

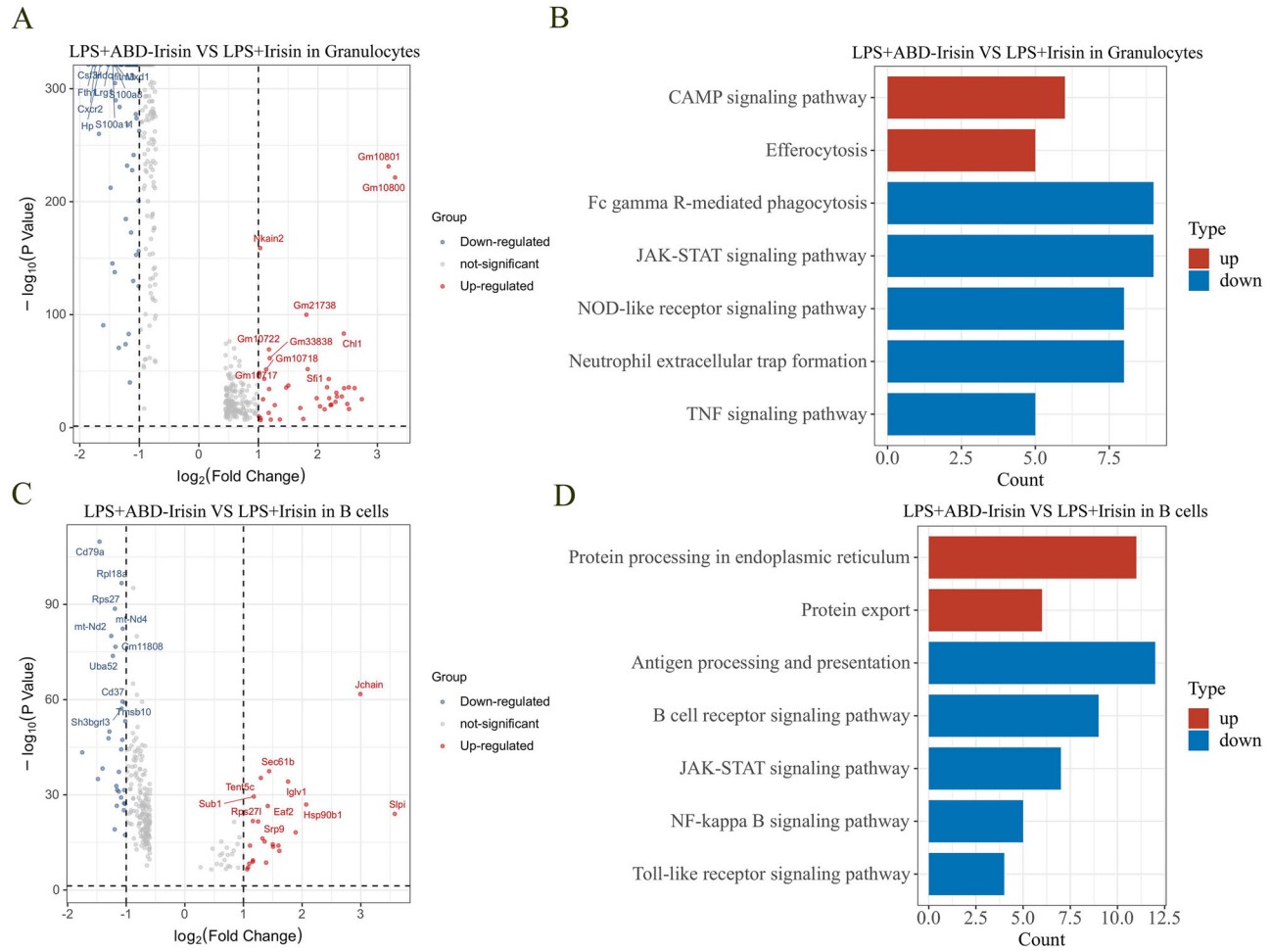

**Fig. 5 | Differential gene expression and KEGG signaling pathways in granulocytes and B cells: LPS + ABD-Irisin *vs* LPS+Irisin. A** DEGs in granulocytes. **B** Enriched KEGG pathways in granulocytes. **C** DEGs in B cells. **D** Enriched KEGG pathways in B cells.

adipocytes[5]. Furthermore, Irisin promotes adaptive thermogenesis *via* activation of PPARα and modulation of the PI3K/AKT and P38MAPK/ERK pathways, facilitating the conversion of white adipose tissue to brown adipose tissue[21]. Our results confirm that both Irisin and ABD-Irisin significantly reduced lipid accumulation in adipocytes, validating the biological activity of both proteins. Co-IP assays further confirmed that the ABD-Irisin fusion retains functional ABD domains capable of binding human serum albumin in vitro.

In LPS-induced inflammatory models, ABD-Irisin treatment demonstrated superior anti-inflammatory efficacy compared to native Irisin. ABD-Irisin markedly reduced plasma IL-6 and TNF-α levels in treated mice, exhibiting greater potency in suppressing IL-6 at equivalent doses. Additionally, ABD-Irisin significantly ameliorated LPS-induced splenomegaly and histopathological inflammation. scRNA-seq provided further mechanistic insight, revealing that ABD-Irisin effectively mitigates systemic inflammation by modulating key inflammatory pathways at the cellular level.

The immunomodulatory role of Irisin in inflammation remains relatively unexplored. The bone marrow, as a primary immune organ, coordinates hematopoiesis and immune cell differentiation[22]. Under systemic inflammatory conditions, hematopoietic stem and progenitor cells (HSPCs) respond to pathogen-derived signals and cytokines, shifting towards myeloid-biased differentiation[23–25]. Our scRNA-seq analysis demonstrated that ABD-Irisin reduced myeloid cell and granulocyte populations while promoting lymphoid and B cell populations, suggesting its potential to restore immune homeostasis. ABD-Irisin notably downregulated

inflammation-related pathways, including antigen presentation, JAK-STAT, NF-κB, and TLR signaling pathways in B cells, further supporting its robust anti-inflammatory effects[26].

Mechanistically, both Irisin and ABD-Irisin suppressed activation of the TLR4-MyD88 signaling pathway by inhibiting the expression of *Lbp* and *Cd14*, thereby attenuating LPS recognition and downstream *MyD88* activation[27]. Additionally, significant inhibition of the NF-κB pathway was observed, likely mediated by upregulated *IκB* expression. Notably, ABD-Irisin demonstrated stronger inhibition of TLR4-MyD88 gene expression and subsequent NF-κB pathway activation, possibly due to intrinsic anti-inflammatory properties conferred by albumin[28]. Albumin has previously been shown to inhibit *TNF-α*-induced *VCAM-1* expression, *NF-κB* activation, and to reduce *IFN-γ* and *TNF-α* expression in a dose-dependent manner[29]. Clinical studies also support an inverse correlation between serum albumin levels and pro-inflammatory cytokines such as IL-6 and TNF-α[30]. Moreover, the higher tissue and plasma concentrations achieved with ABD-Irisin likely amplify its ability to suppress key inflammatory mediators.

Despite these promising results, there are some limitations that needs to be mentioned for current study. For example, splenomegaly assessment relied solely on organ length measurements, and did not include weight or volume measurements to calculate spleen indices. Furthermore, we did not evaluate the MD2 component of the TLR4-MD2 complex, whether Irisin directly interfered with the formation of the LPS-TLR4-MD2 complex remains unknown. Although ABD-Irisin significantly enhanced systemic exposure and anti-inflammatory efficacy, its effects on proteins expression

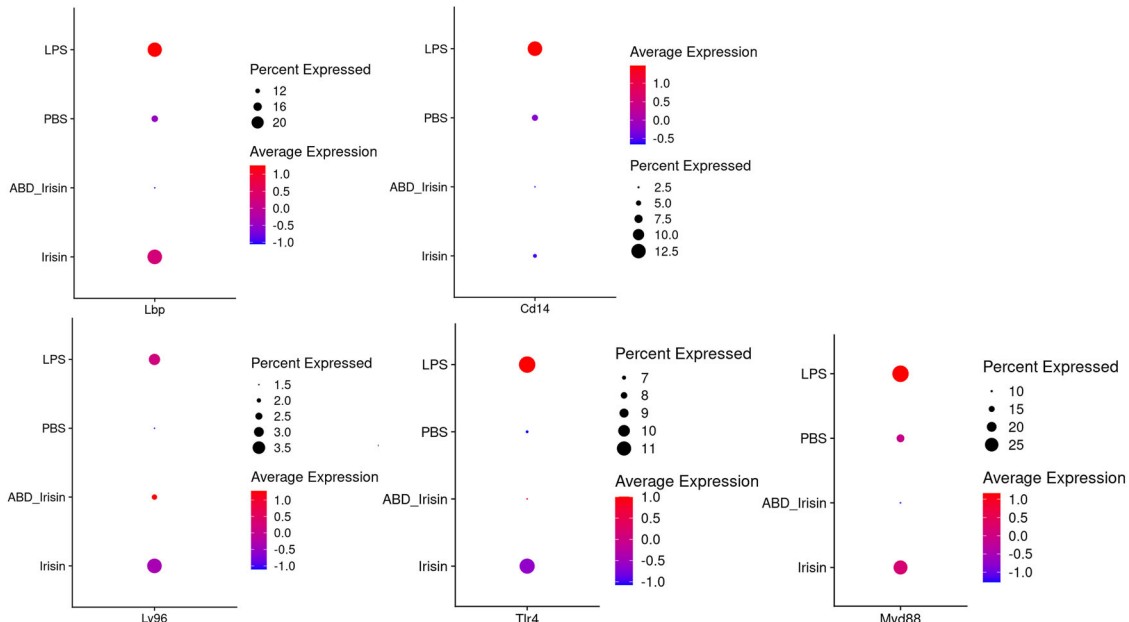

**Fig. 6 | Expression of TLR4-MD2 pathway-related genes in different treatment groups.** Relative mRNA expression levels of key components of the TLR4-MD2 signaling pathway were analyzed across treatment groups. Dot plots display gene expression, with dot size indicating the percentage of cells expressing a given gene and dot color intensity representing the average expression level within each cell group. *Lbp, Cd14, Ly96, Tlr4,* and *Myd88* expression were attenuated in the Irisin group and markedly reduced in the ABD-Irisin group compared to LPS treated group.

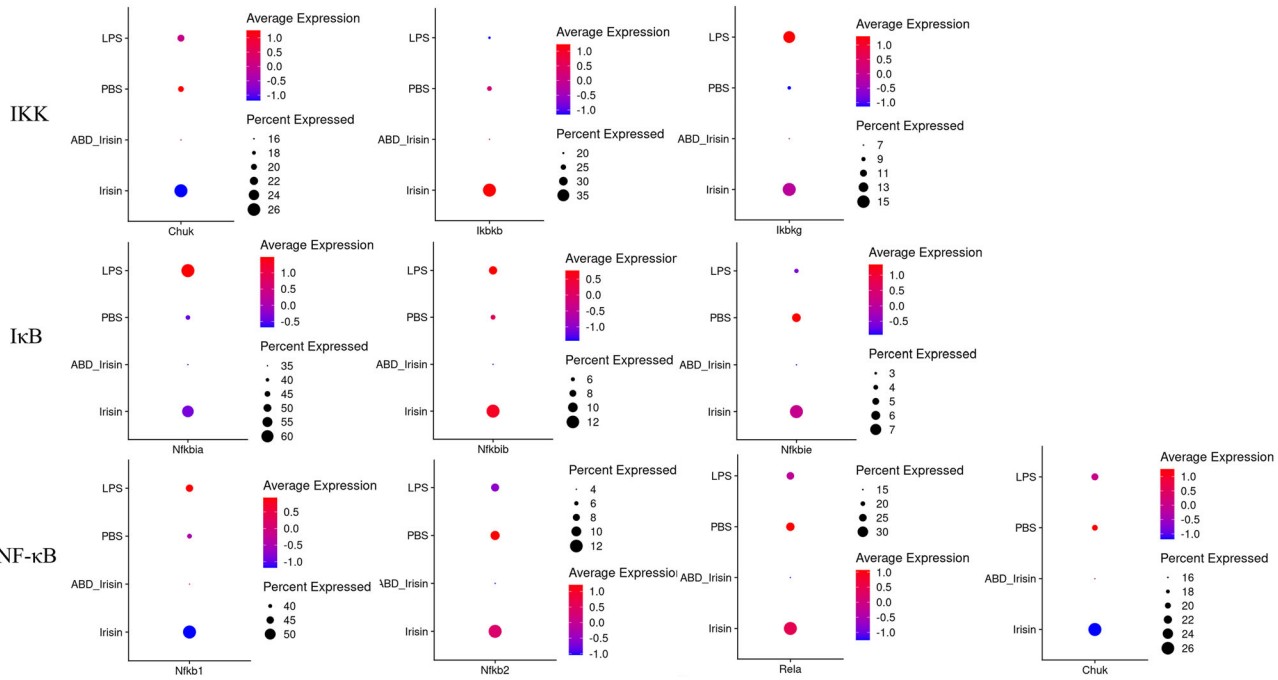

**Fig. 7 | Expression of NF-κB pathway-related genes in different treatment groups.** Relative mRNA expression levels of NF-κB signaling-related genes were assessed across different groups. Dot plots display gene expression, with dot size indicating the percentage of cells expressing a given gene and dot color intensity representing the average expression level within each cell group. Genes including *Ikbkg, Nfkbia, Nfkb1,* and *Chuk* showed attenuated expression in the Irisin group and were further reduced in the ABD-Irisin group compared to LPS treated group.

in TLR4-MyD88 and NF-κB pathway at the cellular level were comparable to native Irisin. This suggested that while extended half-life improves bioavailability, it might not amplify intrinsic target engagement per molecule. Future studies should explore lower doses of ABD-Irisin to determine if equivalent pathway suppression can be achieved with reduced dosing frequency.

In conclusion, ABD-Irisin effectively enhances Irisin's half-life and anti-inflammatory potency, demonstrated by reductions in inflammatory cytokines, tissue damage, and pathway modulation in plasma, spleen, and bone marrow. These findings highlight ABD-Irisin's therapeutic potential, providing a foundation for further research and development of optimized anti-inflammatory treatments.

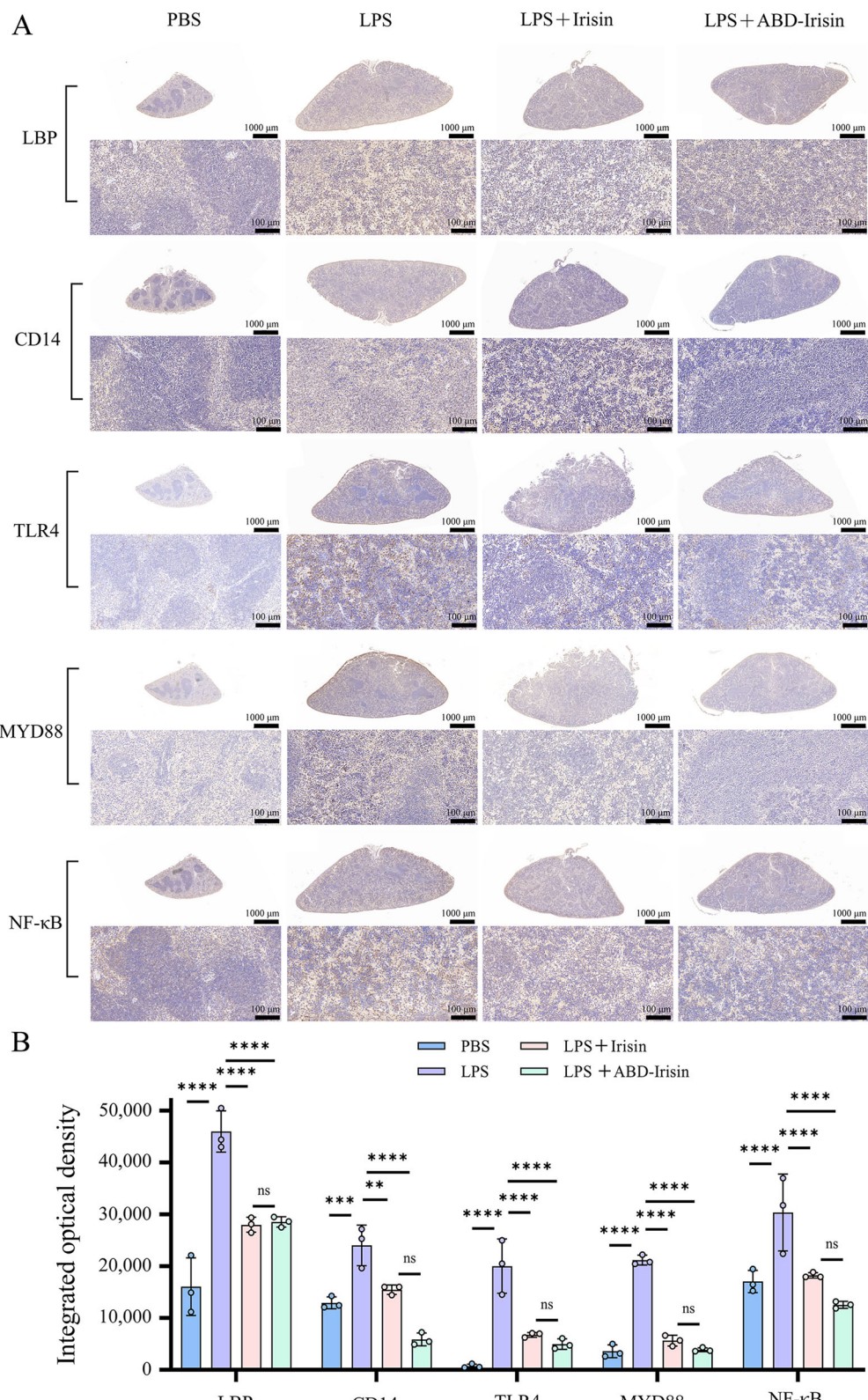

**Fig. 8 | Immunohistochemical staining of *Lbp, Cd14, Tlr4, Myd88,* and *NF-κB* in mice from different treatment groups. A** Immunohistochemical staining images. **B** Differences in cumulative optical density of immunohistochemical staining among treatment groups (n = 3 per group). Data are presented as mean ± S.D. Statistical significance: *$P < 0.05$, **$P < 0.01$, ***$P < 0.001$, ****$P < 0.0001$, ns: not significant.

## Methods

### Materials

HEK-293T and HEK-293F cell lines were obtained from the Precision Medicine Laboratory cell bank at Zhengzhou University. The 3T3-L1 cell line was purchased from Cyagen Biotechnology Co., Ltd. (Guangzhou, Catalog #: M5-0101). Male C57BL/6 J mice (n = 65, 6 weeks old) were purchased from Zhejiang Vital River Laboratory Animal Technology Co., Ltd. (Pinghu, China) and maintained under specific pathogen-free (SPF) conditions at the Experimental Animal Center of Zhengzhou University. The Plenti-3×Flag-Irisin, Plenti-3×Flag-ABD-Irisin plasmids, and E. coli Stable competent cells were constructed by the Precision Medicine Laboratory of Zhengzhou University. All animal experiments were approved by the Ethics Committee of Zhengzhou University Laboratory Animal Center (approval number: ZZU-LAC20230825[03]). Mice were euthanized by carbon dioxide inhalation, consistent with the recommendations of the American Veterinary Medical Association (AVMA) Panel on Euthanasia. We have complied with all relevant ethical regulations for animal use.

### Fusion protein construction

The Irisin sequence was engineered with an upstream signal peptide to facilitate extracellular secretion and preceded by a Kozak sequence to ensure optimal translation initiation. A His-tag was added to the N-terminus, separated from Irisin by a TEV cleavage site. The Irisin sequence was codon-optimized for mammalian cell expression and cloned into the Plenti-3×Flag (P3F) vector (Fig. S1A). For ABD-Irisin construction, an ABD sequence connected by a triple glycine-serine (3×GS) linker was inserted between the TEV cleavage site and Irisin sequence (Fig. S1B). Synthesized plasmids were transformed into E. coli Stable competent cells and cultured on LB agar at 30 °C for 12–15 h. Single colonies were inoculated into LB broth containing 100 μg/mL ampicillin (Beyotime, Shanghai, Catalog #: ST008)and cultured at 30 °C with shaking (300 rpm) for 14 h. Plasmids were isolated using the TIANprep Mini Plasmid Kit (TIANGEN, Beijing, Catalog #: DP103) and verified by Sanger sequencing.

### Protein expression and purification

Fusion protein expression and purification followed previously established protocols[31]. Briefly, HEK-293T cells were seeded into 6-well plates (1 × 10^6 cells/well) 24 h before transfection. At 80–90% confluence, the medium was replaced with serum-free DMEM (Gibco, Catalog #: 11995500). Cells were transfected with a mixture of ABD-Irisin plasmid (1.2 μg), psPAX2 (0.9 μg), pMD2.G (0.9 μg), and polyethyleneimine 25000 (PEI25000 at 1 mg/mL, 25 μL) in serum-free medium. After incubation for another 72 h, supernatants were centrifuged and filtered with 0.22 μm sterile filter and further used for transfection.

HEK-293F cells were cultured in Union 293 medium (Union-Biotech, Shanghai, Catalog #: C0351) at 37 °C, 8% CO_2, 125 rpm. Cells were diluted to 5 × 10^5 cells/mL and infected with lentivirus packaging solution (500 μL) mixed with polybrene (5 μL) (Beyotime, Shanghai, Catalog #: PI470). Cell culture medium was refreshed after 8 h. Puromycin (Thermo Fisher, Catalog #: A1113803) selection (2 ng/mL, increasing to 5 ng/mL gradually) was initiated at 92 h post-infection. Cells were expanded until reaching over 90% viability at 5 ng/mL puromycin selection pressure. Supernatants were collected and underwent buffer exchange into PBS via hollow fiber ultra-filtration and were purified using Ni-NTA affinity chromatography followed by size-exclusion chromatography. Purified proteins were analyzed using SDS-PAGE and Western blotting. All antibodies with source are listed in supplementary Table S3.

### ABD-Irisin structure and activity validation

In pull-down assays, 20 μL Ni-NTA magnetic beads were incubated with 1 mL Irisin or ABD-Irisin (100 ng/mL in PBST) at 4 °C for 2 h to immobilize the proteins. Ni-NTA beads were then immobilized on magnetic separation rack and supernatants were discarded. Ni-NTA beads were incubated with HSA (GlpBio, Catalog #: GP24392) (100 ng/mL) for another 6 h. Finally, proteins were eluted using SDS-PAGE loading buffer (YamayBio, Shanghai, Catalog #: PG212) (95 °C, 5 min) and analyzed via Western blotting.

3T3-L1 preadipocytes were differentiated into mature adipocytes using DMEM containing 0.5 mM IBMX, 0.25 μM dexamethasone, and 10 mg/L insulin. Mature adipocytes were treated with Irisin and ABD-Irisin (0-400 nM) for 96 h. Lipid accumulation was quantified using Oil Red O (BASO, Zhuhai, Catalog #: BA4081) staining followed by extraction with isopropanol (Fuyu Chemical, Tianjin, Catalog #: 67-63-0) and absorbance measurements.

### Pharmacokinetic experiment

Irisin and ABD-Irisin were administered intravenously to male C57BL/6 mice at doses of 100, 500, and 1000 μg/kg (n = 4 per group). Blood samples (50 μL) were collected from the retro-orbital venous plexus at 0.5, 1, 2, 4, 8, 12, and 24 h post-injection. Plasma Irisin concentrations were quantified using the Human/Rat/Mouse Irisin ELISA Kit (Beyotime, Shanghai, Catalog #: PI470).

### H&E staining and immunohistochemistry (IHC)

H&E staining and IHC was performed on major organs (heart, liver, spleen, lung, and kidney) of mice from the PBS, LPS, LPS + Irisin (500 μg/kg), and LPS + ABD-Irisin (500 μg/kg) groups. Samples from the five tissues were fixed using 4% paraformaldehyde for 24 h at room temperature, and embedded using paraffin. Paraffin embedded specimens were sliced, deparaffinized, rehydrated, and stained with H&E. The immunohistochemistry staining followed the protocols described in a previous study[32]. The quantification results for the protein expression on the immunohistochemistry slides were analyzed using the ImageJ software (National Institutes of Health, Bethesda, MD, USA).

### Anti-inflammatory effect of ABD-Irisin

Systemic inflammation was induced in C57BL/6 mice via intraperitoneal injection of lipopolysaccharide (LPS, 1000 ng/g body weight, 100 μL) (Merck, Catalog #: 93572-42-0). Mice (n = 40) were randomly assigned to 8 groups (PBS, LPS, LPS+Irisin 100 μg/kg, LPS+Irisin 500 μg/kg, LPS+Irisin 1000 μg/kg, LPS + ABD-Irisin 100 μg/kg, LPS + ABD-Irisin 500 μg/kg, LPS + ABD-Irisin 1000 μg/kg), with 5 mice per group. Fusion proteins were administered intravenously following LPS injection. Plasma IL-6 and TNF-α levels were measured using Mouse IL-6 ELISA Kit (Elabscience, Catalog #: E-EL-M0044) and Mouse TNF-α ELISA Kit (Elabscience, Catalog #: E-EL-M3063), and body weights were monitored daily.

### Single-cell RNA sequencing

Bone marrows were washed by PBS from mice humerus, ulna, femur, tibia. Cells were centrifuged at 500 g for 5 min. Then the supernatants were discarded, and 3-4 mL red blood lysis buffer was added to lysis red blood cells. After 5 min, 30-40 mL PBS were added to the tube to end the reaction. Cells were resuspended and filtered with 70 μm and 40 μm cell strainers. Cells were the centrifuged and supernatants were discarded. Appropriate amount of PBS (Mg-/Ca-) were added to resuspend the cell pellet. Then cells were counted, and cell viability was assessed to make appropriate concentration (1000–1500 cells/μL). Single-cell suspensions were processed using BGI's DNBelab C4 single-cell transcriptome library kit according to the manufacturer's protocol. Cell populations were annotated using lineage-specific markers: myeloid (Itgam, Ly6c, Ly6g, Cd14) and lymphoid (Cd3d, Cd3e, Cd8, Cd79a, Cd79b, Ighd, Nkg7). In subgroup analysis, we mainly focused on genes in TLR4-MyD88 and NF-κB signaling pathway as these two pathways plays critical roles in LPS-induced inflammation.

### Statistics and reproducibility

Statistical analyses were conducted using GraphPad Prism 8.0.1 (San Diego, CA, USA), and the results are shown as mean ± standard deviation (S.D.). Differences among groups were assessed by two-way ANOVA with multiple comparison tests, with significance set at $P < 0.05$. Raw scRNA-seq data were subjected to doublet removal and batch correction, followed by

hierarchical cell annotation using lineage-specific markers (myeloid cells: *Itgam*, *Ly6g*; Lymploid cells: *Cd3d*, *Cd79a*). Subgroup analysis further classified myeloid cells into monocytes/macrophages (*Ccr5*, *Adgre1*) and granulocytes (*Mpo*, *Csf3r*), enabling precise comparison of cell type proportions across groups. DEGs were identified with |logFC| > 0.25 and $P < 0.05$, and KEGG enrichment analyzed inflammatory pathways in LPS-treated *vs* control, Irisin-treated *vs* LPS, and ABD-Irisin-treated *vs* Irisin groups. DotPlots were used to visualize the expression of several key components in TLR4-MyD88 and NF-κB pathways, including *Lbp*, *Cd14*, *Tlr4*. The expression of NF-κB protein in mouse spleen was checked by IHC.

## Reporting summary

Further information on research design is available in the Nature Portfolio Reporting Summary linked to this article.

## Data availability

All data associated with this study are provided within the paper or the Supplemental Information. The source data behind the graphs in the paper can be found in Supplementary Data 1. The fastq files of the scRNA-seq data are available from the SRA database under Project ID: PRJNA1292814. The newly generated plasmids P3F-IRISIN and P3F-ABD-IRISIN have been deposited in the WeKwikGene (Westlake Laboratory Plasmid Repository) under Barcode numbers 0002003 and 0002004, respectively. Any other data is available from the corresponding author upon request.

## Code availability

The custom code used in this study is publicly available at https://github.com/jxshi/ABD-IRISIN-paper.

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

## Acknowledgements

This work was supported by the Medical Science and Technology Research Project of Henan Province (LHGJ20240257); the Project of Basic Research Fund of Henan Institute of Medical and Pharmaceutical Sciences (2024BP0207). The funders had no role in study design, data collection, and interpretation, or the decision to submit the work for publication.

## Author contributions

Conceptualization: J.C.Z., Y.J.Z., X.W.W., L.P., J.X.S., L.Y., and C.X.G.; Methodology: X.J.M., Z.Q., Z.R.K., F.Y.Z., J.C.Z., Y.J.Z., and X.W.W.; Investigation: F.Y.Z., X.Y.Z., M.Y.S., and J.C.Z.; Resources: J.D., Y.S., Y.J.Z., X.W.W., and J.X.S.; Visualization: J.C.Z., Y.J.Z., X.J.M., Z.Q., L.Y., and C.X.G.; Supervision: L.P., F.Y.Z., X.Y.Z., M.Y.S., and J.X.S.; Funding acquisition: J.C.Z. and J.X.S.; Writing—original draft: J.C.Z., Y.J.Z., and X.W.W.; Writing—review and editing: J.C.Z., Y.J.Z., X.W.W., L.P., X.J.M., Z.Q., Z.R.K., F.Y.Z., X.Y.Z., M.Y.S., J.D., Y.S., L.Y., C.X.G., and J.X.S.

## Competing interests

The authors declare no competing interests.
