## [Transparent Peer Review file · Communications Biology]

Engineered Long-Acting Irisin-Albumin Binding Domain Fusion Protein for Enhanced Anti-inflammatory Efficacy in Lipopolysaccharides-induced Systemic Inflammation

Corresponding Author: Dr Jianxiang Shi

Version 0:

Reviewer comments:

Reviewer #1

(Remarks to the Author)

The authors describe the generation of an irisin – fusion protein. By fusing the albumin binding domain (ABD) with irisin, the half-life of the original protein could be significantly prolonged. Moreover, several tests revealed similar or even better functional capabilities of ABD-irisin compared to its unmodified counterpart. Importantly, the convincing in vitro-results were largely confirmed in LPS challenged mice treated with unmodified and ABD-irisin. In summary, they applied successfully an established fusion strategy, shown to be effective with other drugs before and thus, provided the basis for the production of a therapeutically relevant irisin form.

The experiments are straightforward and are presented in a logical manner. There remain only some minor points to be addressed in a revision:

(I) There is no ethical approval available for the animal experiments. Please add the respective permission.

(II) The numbers of animals included in the different studies is not always clear and should be given for each (group of) figures/tables. For instance, 40 mice were used for evaluation of the anti-inflammatory effects of irisin. There were four treatment groups and three concentrations – how many mice were assigned to every group?

(III) The data in tables lack standard deviations and significance markers. Again, how many animals were analyzed per group?

(IV) There are no data provided regarding the specific antibodies and ELISA kits used in this study. This is of special importance as many available antibodies/kits designed for irisin detection are highly unspecific (see e.g. Islam et al. 2021, Maak et al. 2021).

Reviewer #2

(Remarks to the Author)

Thank you for the opportunity to review the manuscript titled " Engineered Long-Acting Irisin-Albumin Binding Domain Fusion Protein for Enhanced Anti-inflammatory Efficacy in Lipopolysaccharides-induced Systemic Inflammation." In this study, the authors developed an innovative albumin-binding domain (ABD)-conjugated Irisin fusion protein to extend its systemic circulation and enhance its therapeutic potential. The findings highlight ABD conjugation as a promising strategy to improve the pharmacokinetics and anti-inflammatory efficacy of Irisin. The study is well-designed, and the manuscript is clearly written, presenting compelling data to support the conclusions.

However, I have a few minor concerns that should be addressed to strengthen the manuscript:

1. TLR4-MYD88-NF- κ B Pathway Effects: While ABD-Irisin reduced the levels of TLR4, MYD88, and NF- κ B component proteins or mRNA, it did not demonstrate superior efficacy compared to unconjugated Irisin in modulating these targets. The authors should clarify whether this lack of superiority is due to experimental limitations, shared mechanisms of action, or other factors, and discuss the implications for the therapeutic advantage of ABD-Irisin.

2. Distinction Between Signaling and Expression: The manuscript refers to changes in the TLR4-MYD88-NF- κ B signaling pathway and the expression levels of TLR4, MYD88, and NF- κ B. These terms appear to be used interchangeably, which may cause confusion. The authors should clearly differentiate between pathway activity and gene/protein expression levels, providing specific evidence for each to avoid ambiguity.

3. Bioinformatic Analysis Details: The bioinformatic analysis of single-cell RNA sequencing data lacks sufficient detail. The

authors should provide a more comprehensive description of the methods, which would enhance the interpretability of the results.

Addressing these concerns will further strengthen the manuscript and improve its clarity and impact. Overall, this is a robust study with significant potential to advance the field of therapeutic protein engineering.

Reviewer #3

(Remarks to the Author)

The manuscript presents a novel and well-designed investigation of a long-acting ABD-Irisin fusion protein that demonstrates enhanced anti-inflammatory efficacy both in vitro and in vivo. The integration of pharmacokinetic enhancement with immune-modulatory effects, supported by scRNA-seq analysis, provides a strong rationale for the translational potential of this engineered molecule.

To the best of my knowledge, based on a comprehensive review of the current literature and patent landscape, this is the first report of fusing Irisin with an albumin-binding domain (ABD) to improve its pharmacokinetic properties and biological activity—representing a novel strategy in the field of therapeutic peptide engineering.

Pending resolution of the minor issues outlined below, I recommend the manuscript for publication.

Issue 1: The manuscript contains redundant and repetitive wording in certain sections (e.g., line 36–38), which may affect clarity. Editing for conciseness is recommended.

Issue 2: Although the Methods section briefly mentions that both Irisin and ABD-Irisin constructs include an N-terminal His-tag (preceding a TEV cleavage site), this important design feature is not clearly restated in the Results section or the Figure 1 legend. As Figure 1C shows that both proteins are detected by the His antibody, it is crucial to explicitly clarify in the text and figure legends that both constructs carry His-tags, to avoid misleading readers into thinking only ABD-Irisin was tagged. Additionally, in Figure 1, the sample lane order is inconsistent between panels B and C: in panel B (Irisin antibody), the order is Marker → Irisin → ABD-Irisin, whereas in panel C (His antibody), it is Marker → ABD-Irisin → Irisin. This inconsistency makes it difficult to directly compare bands across panels and may confuse readers. The lane order should be standardized across related panels for clarity and professional presentation.

Issue 3: Overstatement in describing IHC signal strength in tissues without statistical difference.

The authors state that ABD-Irisin signals were “substantially stronger” in all tissues examined, including heart and spleen (Fig. 3A). However, the quantitative analysis in Fig. 3B shows no statistically significant difference in these two tissues. Therefore, the description in the main text overstates the effect. The authors should revise the statement to reflect the actual statistical outcomes, highlighting only those tissues (liver, lung, kidney) where significant increases were observed.

Issue 4: The authors claim that ABD-Irisin shows “superior efficacy” at 500 µg/kg, but Figures 4B and 4D show no statistical significance between Irisin and ABD-Irisin at this dose. Additionally, Figure 4B lacks statistical annotations (e.g., “ns” or asterisks) for this comparison, further weakening the claim. The statement should be revised to reflect the actual data.

Issue 5: H&E description would benefit from quantification, annotation, and complete figure legend. The authors describe hepatic inflammation as being “significantly mitigated” after treatment, but no quantitative scoring is provided, nor are key histological features annotated in the figure. In addition, Figure 5 lacks a detailed legend, which should include essential experimental information such as treatment dose (e.g., 500 µg/kg) and timing post-injection. To strengthen clarity and interpretation, the authors are encouraged to provide either histological quantification or clear image annotations (e.g., arrows), and to revise the legend for completeness.

Version 1:

Reviewer comments:

Reviewer #2

(Remarks to the Author)

My concerns have been addressed.

Reviewer #3

(Remarks to the Author)

The authors have responded thoroughly to all of the previously raised concerns. Revisions to both the main text and figures have addressed issues of clarity, consistency, and overstatement. Specifically:

- Issue 1 (redundant wording) has been corrected through concise editing.
- Issue 2 (His-tag clarification and figure lane inconsistency) has been fully addressed, with clarifications added to both the Results section and the Figure 1 legend, as well as correction of lane order in Figure 1C.
- Issue 3 (overstatement in IHC results) was revised to align with statistical significance, focusing only on liver, lung, and kidney.
- Issue 4 (claim of superior efficacy at 500 µg/kg) has been corrected in both figure annotation and textual description, with added statistical notations clarifying significance.
- Issue 5 (H&E figure completeness) has been addressed through annotation (arrows), additional experimental detail in the Methods section, and an improved figure legend.

I appreciate the authors' careful attention to these revisions. The manuscript is now clearer, more rigorous, and ready for publication. I have no further concerns.

Point by point response

Journal: Communications Biology

Article title: Engineered Long-Acting Irisin-Albumin Binding Domain Fusion Protein for Enhanced Anti-inflammatory Efficacy in Lipopolysaccharides-induced Systemic Inflammation

Submission ID: COMMSBIO-25-3788A

Responses to the comments from Reviewer #1

1. There is no ethical approval available for the animal experiments. Please add the respective permission.

Response: Thank you for your comment. The ethical approval for animal experiments was provided to the Materials section of the Methods: "All animal experiments were approved by the Ethics Committee of Zhengzhou University Laboratory Animal Center (approval number: ZZU-LAC20230825[03])."

2. The numbers of animals included in the different studies is not always clear and should be given for each (group of) figures/tables. For instance, 40 mice were used for evaluation of the anti-inflammatory effects of irisin. There were four treatment groups and three concentrations – how many mice were assigned to every group?

Response: Thank you for your suggestion. We are sorry for that there are some inaccurate statements. We have revised the manuscript to specify animal numbers for each experiment: For anti-inflammatory effect experiments (Fig. 3): 40 mice were divided into 8 groups (PBS, LPS, LPS+Irisin 100 µg/kg, LPS+Irisin 500 µg/kg, LPS+Irisin 1000 µg/kg, LPS+ABD-Irisin 100 µg/kg, LPS+ABD-Irisin 500 µg/kg, LPS+ABD-Irisin 1000 µg/kg), with 5 mice per group. Hope it should be clearer now.

3. The data in tables lack standard deviations and significance markers. Again, how many animals were analyzed per group?

Response: Thank you for your suggestion. We have added standard deviations (S.D.) and statistical significance markers to all tables (Table S1 and Table S2) in the Supplementary Information. Additionally, the animal count per group has been specified in the table footnotes: Table S1: n = 4 per group; Table S2: n = 5 per group. We believe these additions improve the clarity of the data presentation.

4. There are no data provided regarding the specific antibodies and ELISA kits used in this study. This is of special importance as many available antibodies/kits designed for irisin detection are highly unspecific (see e.g. Islam et al. 2021, Maak et al. 2021).

Response: We appreciate your suggestion. We have added detailed information on the specific antibodies used in

the Supplementary Table S3. Plasma Irisin concentrations were quantified using the Human/Rat/Mouse Irisin ELISA Kit (Beyotime Biotechnology, Shanghai, Catalog #: PI470). Plasma IL-6 and TNF- α levels were measured using Mouse IL-6 ELISA Kit (Elabscience, Catalog #: E-EL-M0044) and Mouse TNF- α ELISA Kit (Elabscience, Catalog #: E-EL-M3063), and body weights were monitored daily.

Supplementary Table S3. List of antibodies

Antibody	Company/Origin	Catalog number
Anti-FNDC5/irisin	Abcam, Waltham, MA, USA	ab174833
Recombinant Anti-His Tag Mouse mAb	ServiceBio Technology Co., Ltd. Wuhan, China	GB151251
Anti-Albumin Rabbit pAb	ServiceBio Technology Co., Ltd. Wuhan, China	GB11319
Anti-LBP Rabbit pAb	ServiceBio Technology Co., Ltd. Wuhan, China	GB113205
Anti-CD14 Mouse mAb	ServiceBio Technology Co., Ltd. Wuhan, China	GB14023
Anti-TLR4 Mouse mAb	ServiceBio Technology Co., Ltd. Wuhan, China	GB12186
Anti-MyD88 Mouse mAb	ServiceBio Technology Co., Ltd. Wuhan, China	GB12269
Anti-NF- κ B p65 Mouse mAb	ServiceBio Technology Co., Ltd. Wuhan, China	GB12997
Anti-IL-1 beta Mouse mAb	ServiceBio Technology Co., Ltd. Wuhan, China	GB122059
Anti-IL-10 Mouse mAb	ServiceBio Technology Co., Ltd. Wuhan, China	GB12108
Goat Anti-Mouse Ig G	HuaBio Co., Ltd., Hangzhou, China	G1006-1
Goat Anti-Rabbit Ig G	HuaBio Co., Ltd., Hangzhou, China	HA1012

Responses to the comments from Reviewer 2

1. TLR4-MYD88-NF- κ B Pathway Effects: While ABD-Irisin reduced the levels of TLR4, MYD88, and NF- κ B component proteins or mRNA, it did not demonstrate superior efficacy compared to unconjugated Irisin in modulating these targets. The authors should clarify whether this lack of superiority is due to experimental limitations, shared mechanisms of action, or other factors, and discuss the implications for the therapeutic advantage of ABD-Irisin.

Response: Thanks for your comment. While ABD-Irisin and native Irisin showed comparable suppression of TLR4-MyD88-NF- κ B expression at the cellular level, this equivalence likely stems from shared intrinsic mechanisms of irisin action rather than experimental limitations. Critically, ABD-Irisin's therapeutic advantage lies in its prolonged pharmacokinetics prolonged pharmacokinetic profile: its extended half-life (approximately 10 h) far exceeds that of native irisin (< 1 h) (Fig. 1). This enables sustained pathway modulation and underpins

ABD-Irisin's superior *in vivo* efficacy—evidenced by significantly reduced systemic cytokines (IL-6/TNF- α ; Fig. 3) and enhanced tissue distribution (Fig. 4). We acknowledge that the lack of dose-sparing effects at the molecular level represents a limitation, potentially requiring further dose optimization (discussed in the Limitations section).

2. Distinction Between Signaling and Expression: The manuscript refers to changes in the TLR4-MYD88-NF- κ B signaling pathway and the expression levels of TLR4, MYD88, and NF- κ B. These terms appear to be used interchangeably, which may cause confusion. The authors should clearly differentiate between pathway activity and gene/protein expression levels, providing specific evidence for each to avoid ambiguity.

Response: We thank the reviewer for this valuable suggestion. We agree that clarifying the difference between gene/protein expression and signaling pathway activity is crucial for precision. The specific modifications have added in the Results section of the revised manuscript: "The TLR4-MyD88 pathway plays a crucial role in mediating LPS-induced inflammation. In current study, LPS upregulated the expression of *Lbp*, *Cd14*, *Ly96*, *Tlr4* and *Myd88*, which are key components in TLR4-MyD88 pathway, while Irisin and ABD-Irisin suppressed both gene expression and downstream signaling, as evidenced by reduced expression of TLR4-MD2 pathway related genes (Fig. 6). Irisin and ABD-Irisin also reversed LPS-induced NF- κ B activation by modulating key regulatory genes, such as *Ikkkg*, *Nfkbia*, *Nfkbib*, *Nfkb1*, *Chuk*, *Ikkkb*, *Nfkbie*, *Nfkb2*, and *Rela*. ABD-Irisin exhibited a more pronounced inhibitory effect (Fig. 7)". In the Discussion section Revised: "ABD-Irisin demonstrated stronger inhibition of TLR4-MyD88 gene expression and subsequent NF- κ B pathway activation, possibly due to intrinsic anti-inflammatory properties conferred by albumin". Hope it should be clearer now.

3. Bioinformatic Analysis Details: The bioinformatic analysis of single-cell RNA sequencing data lacks sufficient detail. The authors should provide a more comprehensive description of the methods, which would enhance the interpretability of the results.

Response: Thanks for your suggestion. This is a good suggestion to improve our manuscript. To address the need for clearer distinction between signaling pathway activity and gene expression, we have expanded the Methods section with detailed bioinformatics analysis steps. The revised content now includes: Raw scRNA-seq data were subjected to doublet removal and batch correction, followed by hierarchical cell annotation using lineage-specific markers (myeloid cells: *Itgam*, *Ly6g*; Lymphoid cells: *Cd3d*, *Cd79a*). Subgroup analysis further classified myeloid cells into monocytes/macrophages (*Ccr5*, *Adgre1*) and granulocytes (*Mpo*, *Csf3r*), enabling precise comparison of cell type proportions across groups. DEGs were identified with $|\log_{2}FC| > 0.25$ and $P < 0.05$, and KEGG enrichment analyzed inflammatory pathways in LPS-treated vs. control, Irisin-treated vs. LPS, and ABD-Irisin-treated vs. Irisin

groups. DotPlots were used to visualize the expression of several key components in TLR4-MyD88 and NF- κ B pathways, including *Lbp*, *Cd14*, *Tlr4*. The expression of NF- κ B protein in mouse spleen was checked by IHC.

Responses to the comments from Reviewer 3

1. The manuscript contains redundant and repetitive wording in certain sections (e.g., line 36–38), which may affect clarity. Editing for conciseness is recommended.

Response: Thank you for pointing out the redundant wording in our manuscript. We are sorry for the mistakes because of our carelessness, and they have been revised carefully in the revised manuscript. Line 36–37: Irisin promotes the browning of subcutaneous white adipose tissue, thereby enhancing energy expenditure and thermogenesis.

2. Although the Methods section briefly mentions that both Irisin and ABD-Irisin constructs include an N-terminal His-tag (preceding a TEV cleavage site), this important design feature is not clearly restated in the Results section or the Figure 1 legend. As Figure 1C shows that both proteins are detected by the His antibody, it is crucial to explicitly clarify in the text and figure legends that both constructs carry His-tags, to avoid misleading readers into thinking only ABD-Irisin was tagged. Additionally, in Figure 1, the sample lane order is inconsistent between panels B and C: in panel B (Irisin antibody), the order is Marker → Irisin → ABD-Irisin, whereas in panel C (His antibody), it is Marker → ABD-Irisin → Irisin. This inconsistency makes it difficult to directly compare bands across panels and may confuse readers. The lane order should be standardized across related panels for clarity and professional presentation.

Response: Thank you for your kind suggestion and comment. We are sorry for that there are some inaccurate statements. The Results section (Expression and Purification) now states: "Irisin and ABD-Irisin proteins were engineered with CD5 signal peptide followed by an N-terminal His-tag separated by a TEV cleavage site (Fig. S1). The proteins were expressed in HEK 293F cells as secreted proteins and supernatant was collected to purify Irisin and ABD-Irisin. Affinity chromatography using Ni-NTA agarose effectively purified Irisin and ABD-Irisin proteins. Both proteins were cleaned with 50 mM imidazole and eluted with 200 mM imidazole (Fig. S2). SDS-PAGE analysis confirmed the molecular weights of Irisin and ABD-Irisin to be approximately 20-32 kDa and 30-32 kDa, respectively (Fig. 1A). Western blot using anti-Irisin and anti-His antibodies verified that protein of interest were successfully expressed and purified (Fig. 1B-C). Identity and purity (>95%) were validated for both proteins". Figure 1 legend was updated to specify: "B: Western blot using anti-Irisin antibody verifying protein integrity. C: Western blot using anti-His antibody confirming N-terminal His-tag in both constructs." The lane order in Figure

1C was revised. Hope it should be better now.

3. Overstatement in describing IHC signal strength in tissues without statistical difference. The authors state that ABD-Irisin signals were “substantially stronger” in all tissues examined, including heart and spleen (Fig. 3A). However, the quantitative analysis in Fig. 3B shows no statistically significant difference in these two tissues. Therefore, the description in the main text overstates the effect. The authors should revise the statement to reflect the actual statistical outcomes, highlighting only those tissues (liver, lung, kidney) where significant increases were observed.

Response: Thanks for your suggestion. We are sorry for that there are some inaccurate statements. The suggested correction has been made. The main text was revised: "Immunohistochemistry conducted 2 h post-injection (500 µg/kg) demonstrated substantially stronger ABD-Irisin signals in liver, lung, and kidney tissues compared to Irisin (Fig. 2A). Irisin accumulated predominantly in lung tissue (Fig. 2B). Notably, ABD-Irisin significantly increased tissue concentrations of Irisin in liver, lung, and kidney compared to native Irisin (Fig. 2B)". Hope it should be better now.

4. The authors claim that ABD-Irisin shows “superior efficacy” at 500 µg/kg, but Figures 4B and 4D show no statistical significance between Irisin and ABD-Irisin at this dose. Additionally, Figure 4B lacks statistical annotations (e.g., “ns” or asterisks) for this comparison, further weakening the claim. The statement should be revised to reflect the actual data.

Response: Thanks for your suggestion. We are sorry for the mistakes because of our carelessness, and they have been revised carefully in the revised manuscript. We added “asterisks” annotations to Figure 3B (revised version) explicitly showing significant IL-6 reduction for ABD-Irisin vs. Irisin at 500 µg/kg ($P < 0.05$). The sentence in Results section has been updated to: "ABD-Irisin (500 µg/kg) showed significantly stronger IL-6 suppression than native Irisin ($P < 0.05$), though TNF- α reduction was comparable at this dose". In addition, the sentence in the Abstract section was also revised as follows: "In a lipopolysaccharide (LPS)-induced mouse model of systemic inflammation, both Irisin and ABD-Irisin significantly reduced plasma TNF- α levels, splenomegaly, and histopathological inflammation. Notably, ABD-Irisin (500 µg/kg) demonstrated significantly enhanced suppression of plasma IL-6 and splenic inflammatory cytokines (IL-1 β , IL-10) compared to native Irisin". Hope it should be clearer now.

5. H&E description would benefit from quantification, annotation, and complete figure legend. The authors describe hepatic inflammation as being “significantly mitigated” after treatment, but no quantitative scoring is provided, nor are key histological features annotated in the figure. In addition, Figure 5 lacks a detailed

legend, which should include essential experimental information such as treatment dose (e.g., 500 µg/kg) and timing post-injection. To strengthen clarity and interpretation, the authors are encouraged to provide either histological quantification or clear image annotations (e.g., arrows), and to revise the legend for completeness.

Response: We sincerely appreciate the reviewer's valuable suggestions to enhance the clarity of our H&E results presentation. H&E staining was performed on major organs (heart, liver, spleen, lung, and kidney) of mice from the PBS, LPS, LPS+Irisin (500 µg/kg), and LPS+ABD-Irisin (500 µg/kg) groups. Samples from the five tissues were fixed using 4% paraformaldehyde for 24 h at room temperature, and embedded using paraffin. Paraffin embedded specimens were sliced, deparaffinized, rehydrated, and stained with H&E. The quantification results for the protein expression on the immunohistochemistry slides were analyzed using the ImageJ software (National Institutes of Health, Bethesda, MD, USA). The essential experimental information details of H&E staining are described in the Methods section. In addition, we have now annotated the key histological feature (inflammatory cell infiltration) in the representative liver H&E images of the LPS group using blue arrow, as specifically suggested by the reviewer. The arrow clearly pinpoints the location of inflammation around the central veins in the hepatic lobules (Fig. 4A). The figure legend was updated in the revised manuscript. Hope it should be clearer now.

Response to referees

Journal: Communications Biology

Article title: Engineered Long-Acting Irisin-Albumin Binding Domain Fusion Protein for Enhanced Anti-inflammatory Efficacy in Lipopolysaccharides-induced Systemic Inflammation

Submission ID: COMMSBIO-25-3788A

Response to the Reviewers' Comments

We sincerely thank the editors and reviewers for their time and valuable comments, which have significantly improved our manuscript. We are delighted that the reviewers are satisfied with our previous revisions. Below is our point-by-point response to the final comments.

Reviewer #2 (Remarks to the Author):

Comment: My concerns have been addressed.

Response: We thank the reviewer #2 for the positive feedback and confirmation that all concerns have been adequately addressed.

Reviewer #3 (Remarks to the Author):

Comment: The authors have responded thoroughly to all of the previously raised concerns. Revisions to both the main text and figures have addressed issues of clarity, consistency, and overstatement. Specifically:

- Issue 1 (redundant wording) has been corrected through concise editing.
- Issue 2 (His-tag clarification and figure lane inconsistency) has been fully addressed, with clarifications added to both the Results section and the Figure 1 legend, as well as correction of lane order in Figure 1C.
- Issue 3 (overstatement in IHC results) was revised to align with statistical significance, focusing only on liver, lung, and kidney.
- Issue 4 (claim of superior efficacy at 500 $\mu\text{g}/\text{kg}$) has been corrected in both figure annotation and textual description, with added statistical notations clarifying significance.
- Issue 5 (H&E figure completeness) has been addressed through annotation (arrows), additional experimental detail in the Methods section, and an improved figure legend.

I appreciate the authors' careful attention to these revisions. The manuscript is now clearer, more rigorous, and

ready for publication. I have no further concerns.

Response: We are very grateful to Reviewer #3 for the thorough and constructive assessment of our revisions. We appreciate the confirmation that all raised issues have been resolved to the reviewer's satisfaction and that the manuscript is now ready for publication.